# Changes in Cognitive Functions after Carotid Endarterectomy and Carotid Stenting: A Decade-Apart Comparison

**DOI:** 10.3390/biomedicines12010013

**Published:** 2023-12-20

**Authors:** Daniel Václavík, David Pakizer, Tomáš Hrbáč, Martin Roubec, Václav Procházka, Tomáš Jonszta, Roman Herzig, David Školoudík

**Affiliations:** 1Department of Neurology, University Hospital Ostrava, 708 00 Ostrava, Czech Republic; daniel.vaclavik@vtn.agel.cz (D.V.); martin.r@centrum.cz (M.R.); 2Comprehensive Stroke Centre, Department of Neurology, Charles University Faculty of Medicine and University Hospital, 500 05 Hradec Králové, Czech Republic; herzig.roman@seznam.cz; 3Stroke Centre, Department of Neurology, Hospital Agel Ostrava Vitkovice, 703 00 Ostrava, Czech Republic; 4Centre for Health Research, Faculty of Medicine, University of Ostrava, Syllabova 19, 703 00 Ostrava, Czech Republic; david.pakizer@email.cz; 5Department of Neurosurgery, University Hospital Ostrava, 708 52 Ostrava, Czech Republic; tomas.hrbac@fno.cz; 6Department of Neuroscience, Faculty of Medicine, University of Ostrava, 703 00 Ostrava, Czech Republic; 7Department of Radiodiagnostics, University Hospital Ostrava and Faculty of Medicine, University of Ostrava, 703 00 Ostrava, Czech Republic; vaclav.prochazka@fno.cz (V.P.); tomas.jonszta@fno.cz (T.J.)

**Keywords:** carotid stenosis, carotid endarterectomy

## Abstract

Background: This study investigates changes in cognitive function in patients with severe carotid stenosis who underwent carotid endarterectomy (CEA) and carotid stenting (CAS) over two decades. Methods: We compared cognitive function within 30 days after the procedure in 267 patients (first 100 each for CEA and CAS in two periods: 2008–2012 and 2018–2022) in a single institution. Assessments used Adenbrooke’s Cognitive Examination–Revised (ACE-R), the Mini-Mental State Examination (MMSE), Speech Fluency Test (SFT), and Clock Drawing Test (CDT), conducted before and 30 ± 2 days after surgery. Results: Patients (mean age 67.2 years, 70%+ carotid stenosis) exhibited different cognitive changes over periods. In 2008–2012, significant declines in MMSE (CEA, *p* = 0.049) and CDT (CAS, *p* = 0.015) were observed among asymptomatic patients. On the contrary, in 2018–2022, improvements were observed in ACE-R and MMSE for symptomatic and asymptomatic patients undergoing CEA and CAS. Conclusion: Over a decade, advances in interventional techniques and patient management have reduced risks of cognitive decline in patients with asymptomatic carotid stenosis and also have improved cognitive functions in both symptomatic and asymptomatic individuals.

## 1. Introduction

Internal carotid artery (ICA) stenosis represents the most common cause of ischemic stroke [1], and the stroke risk increases in proportion to the severity of ICA stenosis [2]. According to current recommendations, interventional treatment such as surgical carotid endarterectomy (CEA) or intraluminal carotid angioplasty with stenting (CAS) is indicated for specific patients with ICA stenosis >50% [3,4,5]. Both interventions are considered to be relatively safe and effective with estimated risks of perioperative morbidity and mortality being less than 6% [6,7]. Moreover, perioperative stroke and death rates are <2% for asymptomatic and <4% for symptomatic stenosis cases [8]. However, it is important to note that CAS is still associated with a higher risk of silent cerebral infarctions [9]. According to a recent meta-analysis, diffusion-weighted imaging (DWI) on magnetic resonance (DW-MRI) detected new ischemic cerebral lesions significantly more often after CAS than after CEA, with rates of 40.3% and 12.2% respectively [10]. Furthermore, these brain infarctions, although often asymptomatic at the time of clinical onset, may contribute to an increased risk of future cognitive impairment and dementia [11].

However, the effect of CEA and CAS on cognitive function remains uncertain. The pathophysiological basis for cognitive dysfunction is believed to be microembolization originating from an atherosclerotic plaque (parts of the plaque or attached thrombus) or hypoperfusion caused by arterial stenosis [12,13,14,15,16,17,18,19].

As a result, after the removal of arterial stenosis, an improvement in cognitive functions can be anticipated. In contrast, cognitive deterioration can also occur during the operation due to the exacerbation of hypoperfusion when clamping the artery or as a result of microembolizations during the procedure [10,20,21,22,23].

The review of 13 CAS and CEA studies comparing the outcomes after intervention identified 403 patients after CEA and 368 patients after CAS. In most studies, no changes were observed in cognitive tests. However, a meta-analysis could not be performed due to the limited number of patients with sample sizes ranging from 26 to 129 patients. Furthermore, notable heterogeneity was found in neuropsychological tests conducted, their timing, and the duration of patient follow-up [19]. Maggio et al. also evaluated the impact of new DWI lesions and found a detrimental effect on cognition limited by a small sample size of only 37 patients [24].

The SONOBUSTER study revealed a statistically significant reduction in DWI lesions after CAS and CEA with intraoperative sonolysis. However, the study did not indicate a significant impact of DWI lesions on cognitive functions in either the group with or without the sonolysis. It is worth noting that the study did not have an adequate sample size to reach statistical significance for this particular outcome [25].

Advances in surgical techniques and stenting devices over the past several decades have improved the prognosis of patients with carotid stenosis. In the current study, we examined the progress made over a decade through a retrospective analysis of carotid interventions recorded in a single hospital registry.

The primary objective of our study was to compare the differences in cognitive test outcome after CEA and CAS within individual decades. Secondary objectives were to determine whether CEA and CAS result in cognitive functions alterations and to make a comparison of changes in cognitive tests between patients who underwent CEA and those who underwent CAS.

## 2. Materials and Methods

This retrospective analysis adhered to the principles outlined in the Declaration of Helsinki (1964) and its subsequent amendments, including the most recent one in 2013. All procedures were carried out in accordance with both national and institutional guidelines. The study was approved by the Ethics Committee of the Ostrava University Hospital (approval nos. MZ10-FNO and 497/2017). Data from patients who signed an informed consent to anonymous use of their hospital registry records for scientific purposes were included exclusively in this analysis.

Consecutive patients who underwent CEA or CAS in a two time periods were included to the analysis. The first period was from January 2008 to December 2012 [P1] and the second period from January 2018 to December 2022 [P2]. Inclusion criteria were: (1) ICA stenosis ≥70% detected using duplex sonography, computed tomography angiography or magnetic resonance angiography; (2) indication for CEA or CAS according to valid guidelines [7,26,27,28]; (3) age 40–85 years; signed Informed consent. Exclusion criterion was a failure to perform preoperative and postoperative cognitive testing due to aphasia, dementia, or unwillingness to undergo testing.

To avoid a selection bias, only the first 50 patients who underwent CEA and the first 50 patients who underwent CAS in each selected period and at the same time met the inclusion criteria were selected for the analysis. Finally, 267 out of 400 selected patients from the Ostrava University Hospital underwent cognitive testing and were analyzed.

### 2.1. Carotid Endarterectomy

During the first study observation period (P1), CEA was performed under general anesthesia by an experienced neurosurgeon. A longitudinal skin incision was made along the anterior edge of the sternocleidomastoid muscle to expose the common carotid artery and its two main branches, namely the internal and external carotid arteries. Subsequently, 100 IU of unfractionated heparin per kg of body weight was administered and after 3 min, blood flow was stopped by clamping. The ipsilateral middle cerebral artery was continuously monitored using transcranial Doppler (DWL MultiDop T1, DWL, Sipplingen, Germany) to determine the need for shunt insertion. Then, the longitudinal arteriotomy of the common and internal carotid arteries was performed and plaque was removed. Following plaque extraction, the artery was sutured and the clamp released. Protamine sulfate (1 mL/1000 IU of heparin) was administered 5 min after flow was restored. A thorough homeostasis check was performed, a Redon drain was inserted, and the wound was anatomically closed. The patients were transferred to the intensive care unit, and after 24 h, brain MRI and duplex ultrasound of the ICA were performed.

During the second period (P2), several changes were made to the surgical procedure. Specifically, (i) a transverse skin incision replaced the longitudinal incision, (ii) systemic blood pressure target values were individually predefined based on the mean blood flow velocity in the ipsilateral middle cerebral artery, as monitored by transcranial Doppler, (iii) the anticoagulation strategy was adjusted to 7000 IU heparin preoperatively for patients weighing ≥ 80 kg and 5000 IU for patients weighing < 80 kg, and (iv) protamine sulfate was not administered. The surgical team remained unchanged in both periods.

### 2.2. Carotid Percutaneous Transluminal Angioplasty with Stenting

The patients indicated for CAS underwent diagnostic and interventional procedures using digital subtraction angiography. The procedure was performed through the femoral approach by an experienced interventional radiologist. All patients received long-term aspirin (100 mg/day) and a loading dose of 525-mg of clopidogrel. A dose of 10,000 IU of unfractionated heparin was administered at the beginning of the intervention. Whenever technically feasible, a distal cerebral protection device (FilterWire EZ™; Boston Scientific, Natick, MA, USA) was used during the procedure. Following the insertion of the filter/wire through the area of the stenosis, pre-dilatation of the stenosis was performed as required using a balloon, followed by the insertion of the stent. The selection of the carotid stent and other specific interventional strategies was determined by the judgment and the experience of the interventional radiologist. After predilatation, if necessary, an appropriate stent (such as the Carotid Wall Stent Boston Scientific Corp., Natick, MA, USA) was implanted for each stenosis. Based on angiographic findings, any remaining stenosis within the stent was addressed by further dilation with a balloon, after which the protection device was subsequently removed. Following the procedure, a follow-up angiography was performed and the results were assessed.

In the second period, some changes were made in the procedure protocol. All patients were tested for resistance to acetylsalicylic acid and/or clopidogrel. Furthermore, the dose of unfractionated heparin at the beginning of the intervention was reduced to 5000 IU for patients weighing < 80 kg and 7500 IU for patients weighting ≥ 80 kg. Blood pressure was more precisely controlled through continuous infusion of saline solution. Moreover, advanced instrumentation was used including a new type of double-layered stent (Roadsaver™ Carotid Artery Double-Layer Stent, Terumo Corporation, Tokyo, Japan). The interventional radiology team remained unchanged in both periods.

### 2.3. Clinical Evaluation

Physical and neurological examinations were performed by a board-certified neurologist 24 h and 30 days after the intervention. Neurological deficits were assessed with the National Institutes of Health Stroke Scale (NIHSS), self-sufficiency with the modified Rankin Scale (mRS), and cognitive functions with the Adenbrooke’s Cognitive Examination–Revised (ACE-R), Mini-Mental State Examination (MMSE), Speech Fluency Test (SFT), and Clock Drawing Test (CDT). All tests were performed according to established protocols 12 to 48 h before surgery and 30 ± 2 days after surgery. All patients underwent also brain magnetic resonance imaging 24 h after intervention for detection of new ischemic or hemorrhagic lesions.

### 2.4. Statistical Analysis

The sample size for the study was determined by considering an expected difference of 4 points in the follow-up cognitive test results between intervention in period 1 (estimated 86 ± 5 points) and period 2 (90 points). Calculations indicated that a minimum of 25 patients in each group would be required to detect a significant difference with an α value of 0.05 (two-tailed) and a β value of 0.8.

The normality of the data distributions was tested using the Shapiro–Wilk test. Categorical variables are presented as relative frequencies, while continuous variables are reported as mean ± standard deviation. Continuous variables were compared between groups using the Mann–Whitney U-test, while categorical variables were analyzed using the chi-square test for independence in contingency tables. Changes in cognitive test performance scores were compared using the Wilcoxon matched pairs signed rank test. The significance level was defined as *p* < 0.05 for all tests.

## 3. Results

Of 400 patients who underwent carotid intervention in both study periods, the medical records were examined for 267 patients (181 males, mean age 67.2 ± 7.4 years) with carotid stenosis ≥70% (mean 80.3% ± 8.6%) receiving CEA (93 males and 48 females, mean age 66.6 ± 7.7 years) or CAS (88 males and 38 females, mean age 67.8 ± 7.1 years) during the two study periods (P1 and P2) and completed cognitive tests both before and 30 days after intervention were included in the statistical analysis. All demographic data are summarized in Table 1.

In P1 or P2, no patient with asymptomatic carotid stenosis suffered a stroke, transient ischemic attack (TIA), myocardial infarction, or vascular death within 30 days after intervention. Among patients with symptomatic ICA stenosis who underwent CEA, one patient suffered a stroke in P1 and one patient had TIA in P1. Similarly, in the CAS group, one patient in P1 and one patient in P2 suffered a stroke and one patient in P1 had TIA. Of the 267 patients, only one patient died within 30 days after the intervention due to a stroke. This patient underwent CEA in P2 and the cause of death was a cancer.

During P1, a significant decrease in the MMSE score was observed among asymptomatic patients after CEA (*p* = 0.049), and a similar decline in CDT performance was observed was observed among asymptomatic patients after CAS (*p* = 0.015)—Table 2. In P1, no significant differences were found in other cognitive tests between the two groups. However, during P2, no significant cognitive decline was detected by any of the tests in either group.

During the P2 period, a significant improvement was detected in the ACE-R test (*p* < 0.01) and MMSE (*p* < 0.05) in both symptomatic and asymptomatic patients after CEA. In both symptomatic and asymptomatic patients after CAS, there was an improvement in the ACE-R test scores (*p* < 0.01). Additionally, in symptomatic patients, there was also a significant improvement in the MMSE scores (*p* < 0.05). (Table 2).

Both CEA and CAS resulted in an enhancement of cognitive functions measured by the ACE-R test during the P2 period. However, in terms of CAS, an improvement in MMSE scores was observed exclusively in symptomatic patients.

## 4. Discussion

In a previously published study using the same patient sample, new ischemic lesions in brain diffusion-weighted brain imaging after the procedure were significantly less frequent in the CEA group compared to the CAS group during both P1 (23% *versus* 49%; *p* < 0.001) and P2 (15 *versus* 29 per cent; *p* = 0.017), in the CAS group during P2 compared to P1 (*p* = 0.004), and in the entire cohort during P2 compared to P1 (*p* = 0.002). Significant reductions in the risk of new ischemic lesions in brain DW-MRI follow-up were detected after CAS, with rates decreasing from 46% in asymptomatic and 52% in symptomatic stenosis patients during P1 to 26% and to 32%, respectively, during P2. The mean volume of ischemic lesion was higher in the CEA group compared to the CAS group, although this difference did not reach statistical significance (*p* > 0.050) [29].

In line with the significant impact of new ischemic brain lesions on cognition, the reduction in the risk of new ischemic lesions on the DW-MRI of the brain after the second period was correlated with a reduced risk of cognitive decline. In fact, cognitive functions showed improvement during P2. More specifically, patients exhibited significantly higher ACE-R and MMSE scores 30 days after the procedure compared to their score before the procedure, except for MMSE among asymptomatic patients after CAS. In addition to the expected gradual restoration of cognitive function after stroke among patients with symptomatic carotid stenosis, which can be attributed to neuroplastic changes, the observed cognitive improvement may stem from better cerebral hemodynamics after CEA and CAS. In fact, cerebral blood supply dysfunction can be considered an independent and potentially reversible factor that plays a role in determining cognitive decline in patients with severe carotid stenosis [30] Although many studies have reported an improvement in patient cognitive functions after CAS [31,32,33,34,35,36] and CEA [31,37,38], these benefits appear to be restricted to symptomatic patients, presumably with greater disruption of blood flow. For example, a recent systematic review concluded that neither CAS nor CEA consistently improve overall cognitive function in asymptomatic patients (<2%). This is especially the case for patients who are otherwise at low risk of cognitive decline, even over the long term with appropriate medical treatment [39]. If transfemoral approach represents a high risk for patient, transcervical CAS seems to be an option with good clinical outcome and reduction of postoperative DW-MRI ischemic lesions related with filter-protected CAS [40]. Compared to interventional treatment, medical treatment does not show improvement in cognitive performance in the patient after CAS or CEA in one year of follow-up [41].

Due to its high spatial resolution, DW-MRI is commonly used for the detection of small brain infarcts after CEA and CAS [42,43]. Furthermore, cerebral infarcts detected with DW-MRI are often the primary endpoint in recent trials instead of 30-day vascular morbidity and mortality. This change is due to a considerably higher incidence of cerebral infarcts compared to clinically symptomatic and potentially fatal vascular events, such as transient ischemic attack (TIA) or stroke [44,45]. In addition, these infarcts are associated with a higher risk of cognitive decline or dementia [11,46,47,48,49,50,51]. Our previous results [29] are consistent with a recently published systematic review [9] and a meta-analysis [52]. They all conclude that new ischemic lesions detected on brain DW-MRI are more common after CAS than after CEA, with an incidence ranging from 18% to 58% [53].

In the second period, there was a significant reduction in the risk of new ischemic lesions detected on the follow up DW-MRI [29]. One possible explanation for this improvement is the greater experience of the CEA and CAS teams. However, interventional procedures were also modified, including more precise intraoperative blood pressure management, as well as increased use of antiplatelet treatment or heparin management, and perhaps even a greater improvement in primary and secondary stroke prevention in the Czech Republic [54]. A recent study found that continuous TCD monitoring reduced the risk of new ischemic lesions detected on DW-MRI after CAS or CEA [25], while neither the type of distal protection [55] nor stent design [56] exhibited similar effects. Furthermore, a radiological study found that the majority of microemboli (particularly the smaller and more distal ones) did not show any manifestation [57,58] or disappeared during long-term MRI follow-up [59].

We observed a tendency for smaller ischemic lesions after CAS compared to CEA, although this difference did not reach statistical significance [29]. This observation contrasts with two previous studies that found similar volumes of lesion after both procedures [43,60]. One possible explanation for this discrepancy is that during CAS atherosclerotic debris and dislodged thrombotic material could result in multiple small emboli, each affecting only a small volume of brain tissue, whereas CEA may generate fewer emboli, but could involve a larger volume of brain tissue.

### Study Limitations

There are some limitations to this study. First, the retrospective single-center study design may have introduced selection bias. Thus, the results may not be universally representative of the safety comparison between CAS and CEA. On the contrary, the single-center design allows for a direct and unambiguous comparison between the two methods over a decade, as both CAS and CEA intervention teams remain consistent. Secondly, cognitive tests were performed only one month after the procedure. For a more comprehensive assessment of definitive cognitive changes, a prospective study with a one-year follow-up would be appropriate.

## 5. Conclusions

Our study showed that improvements in CEA and CAS management, along with a reduction in the frequency of microembolizations, contributed to the prevention of cognitive decline after the procedure in asymptomatic patients. Furthermore, these improvements led to improved cognitive functions in both symptomatic and asymptomatic patients.

## Figures and Tables

**Table 1 biomedicines-12-00013-t001:** Demographic data.

	CEA Period 1		CEA Period 2		CAS Period 1		CAS Period 2	
	AS	SS	AS	SS	AS	SS	AS	SS
Number of participants; *n*	35	32	35	39	33	28	36	29
Age (years); mean ± SD; years	65.7 ± 8.5	65.9 ± 6.9	67.6 ± 6.8	67.1 ± 8.7	66.4 ± 7.6	67.1 ± 7.3	68.9 ± 6.7	68.6 ± 6.9
Male sex; *n* (%)	23 (66)	21 (66)	24 (69)	25 (64)	22 (67)	20 (71)	25 (70)	21 (72)
Right-sided stenosis; *n* (%)	20 (57)	17 (53)	19 (54)	22 (56)	17 (52)	14 (50)	19 (53)	16 (55)
Severity of stenosis; mean ± SD; %	80.1 ± 8.1	80.9 ± 8.7	79.2 ± 7.9	79.8 ± 9.2	81.1 ± 9.6	80.3 ± 8.9	80.1 ± 7.7	80.6 ± 8.7
Contralateral stenosis >50%; *n* (%)	12 (34)	10 (31)	11 (31)	13 (33)	15 (45)	11 (39)	14 (39)	11 (38)
Arterial hypertension; *n* (%)	29 (83)	28 (88)	31 (89)	33 (85)	30 (91)	25 (89)	34 (94)	26 (90)
Diabetes mellitus; *n* (%)	11 (31)	11 (34)	11 (31)	13 (38)	13 (39)	12 (43)	13 (36)	10 (34)
Hyperlipidemia; *n* (%)	25 (71)	23 (72)	26 (74)	30 (77)	22 (67)	21 (75)	26 (72)	22 (76)
Coronary heart disease; *n* (%)	10 (29)	11 (34)	12 (34)	11 (28)	13 (39)	11 (39)	13 (36)	9 (31)
Atrial fibrillation; *n* (%)	2 (6)	2 (6)	1 (3)	2 (5)	2 (6)	1 (4)	1 (3)	2 (7)
Smoking; *n* (%)	11 (31)	10 (31)	12 (34)	12 (31)	9 (27)	8 (29)	13 (36)	10 (34)
Alcohol abuse; *n* (%)	1 (3)	0 (0)	2 (6)	2 (5)	2 (6)	1 (4)	0 (0)	2 (6)
Statin use; *n* (%)	24 (69)	21 (66)	26 (74)	30 (77)	21 (64)	21 (75)	26 (72)	22 (76)
Antiplatelets; *n* (%)	32 (91)	30 (94)	34 (97)	38 (97)	33 (100)	28 (100)	36 (100)	29 (100)

AS—asymptomatic stenosis; CAS—carotid angioplasty and stenting; CEA—carotid endarterectomy; SD—standard deviation; SS—symptomatic stenosis, No significant differences were observed in any demographic data between the groups.

**Table 2 biomedicines-12-00013-t002:** Results of cognitive tests prior to and 30 days after intervention.

	BS	AS	Δ	BS	AS	Δ	BS	AS	Δ	BS	AS	Δ
	**CEA Period 1** **AS (*n* = 35 s)**			**CEA Period 1** ** SS (*n* = 32)**			**CEA Period 2** **AS (*n* = 35)**			**CEA Period 2** **SS (*n* = 39)**		
ACER; median (IQR); mean	87 (85.5–91.5); 87.2	87 (84.5–91); 86.9	−1 (−1.5–0.5); −0.3	83.5 (70–89); 79.9	83.5 (69–89); 78.8	−0.5 (−3–0); −1.1	87 (77–91); 84.0	91 (82–95.5); 88.0	3 (1.5–7); 4.0 ^#^	83 (72.5–89); 80.6	89 (81.5–91.5); 86.5	6 (2–8); 5.9 ^#^
MMSE; median (IQR); mean	29 (28–29.5); 28.6	28 (27–29); 27.9	0 (−1.5–0); −0.7 *	28 (25–29); 26.6	27.5 (25–29); 26.1	0 (−1–0); −0.5	28 (26–28.5); 27.0	29 (26–30); 27.8	1 (0–2); 0.8 *	27 (24–29); 26.1	28 (27–29); 27.6	1 (0–3); 1.5 *
Clock Drawing Test; median (IQR); mean	5 (4–5); 4.1	5 (4–5); 3.9	0 (0–0); −0.2	4 (1–5); 3	4 (0–5); 2.8	0 (0–0); −0.2	5 (4–5); 4.5	5 (4.5–5); 4.6	0 (0–0); 0.1	5 (4–5); 4.3	5 (4–5); 4.6	0 (0–0.5); 0.2
Speech Fluency Test; median (IQR); mean	10 (7.5–11.5); 9.4	10 (8–11); 9.5	0 (0–1); 0.1	6.5 (4–10); 7.1	7.5 (5–10); 7.1	0 (−1–1); 0	10 (8–12); 9.4	10 (9–12.5); 9.9	0 (−1–1); 0.5	10 (7–12.5); 9.3	10 (7–12); 9.5	0 (−1–1); 0.2
	**CAS period 1** **asymptomatic (33 patients)**			**CAS period 1** **symptomatic (28 patients)**			**CAS period 2** **asymptomatic (36 patients)**			**CAS period 2** **symptomatic (29 patients)**		
ACER; median (IQR); mean	87 (77–93); 85.0	88 (76–93); 83.9	0 (−3–1); −1.1	83 (71–90.5); 81.1	82 (68–91.5); 78.6	1 (−2–2); −2.5	87 (76–92); 83.5	89 (83–94); 86.3	3 (1–5); 2.8 ^#^	83.5 (80–87); 81.7	88 (84–94); 85.3	3 (2–6); 3.6 ^#^
MMSE; median (IQR); mean	28 (26–29); 26.9	28 (25–28); 26.8	0 (−1–1); −0.2	26.5 (25–29); 26.1	27 (25–28); 25.2	0 (0–1); −0.9	28 (25–29); 26.9	28.5 (26–29); 27.4	1 (−1–1); 0.5	28 (25–29); 26.8	28 (27–30); 27.8	1 (0–2); 1.0 *
Clock Drawing Test; median (IQR); mean	4 (2–5); 3.2	3 (0–5); 2.6	0 (−1–0); −0.6 *	4 (0–5); 2.9	4 (0–5); 2.8	0 (0–0); −0.1	5 (4–5); 4.5	5 (4–5); 4.5	0 (0–0); 0	5 (4–5); 4.3	5 (4–5); 4.5	0 (0–1); 0.2
Speech Fluency Test; median (IQR); mean	10 (6–12); 9.1	10 (5–12); 8.7	0 (−1–1); −0.4	10 (4–11); 8.0	8.5 (5–11); 8.4	0 (−0,5–1); −0.4	10 (8–11); 9.7	10 (8–12); 9.8	0 (−1–2); 0.1	10 (8–12); 9.6	10 (9–12); 9.6	0 (0–1); 0.0

*—*p* < 0.05; ^#^—*p* < 0.01; ACER—Adenbrook’s Cognitive Examination—Revised; CAS—carotid angioplasty and stenting; CEA—carotid endarterectomy; IQR—interquartile range; MMSE—Mini Mental State Examination.

## Data Availability

All source data are available from the corresponding author upon request.

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
