# Peer review of "Changes in Cognitive Functions after Carotid Endarterectomy and Carotid Stenting: A Decade-Apart Comparison"

_biomedicines, 2023, doi:10.3390/biomedicines12010013_

Round 1
Reviewer 1 Report
Comments and Suggestions for Authors
I read with great attention and interest the aper entitled "Changes in cognitive functions after carotid endarterectomy and carotid stenting: a decade-apart comparison."
I think the paper is interesting and welle reported.
I have only some suggestions for the Auhtors:
- please provide a more descriptive methods section, better reporting how patients were enrolled during the different phases of the study; in tis present form M&M section is quite difficult to interoretate about this specific point
- linguist revision (sse below) is mandatory
- please considere to discuss the following paper: Faraglia V, Palombo G, Stella N, Rizzo L, Taurino M, Bozzao A. Cerebral embolization during transcervical carotid stenting with flow reversal: a diffusion-weighted magnetic resonance study. Ann Vasc Surg. 2009 Jul-Aug;23(4):429-35. doi: 10.1016/j.avsg.2008.09.009. Epub 2008 Nov 28. PMID: 19041221.
Comments on the Quality of English LanguageThe whole manuscript requires an extensive linguistic revision, such as some sentences (like the following: Out of the 267 patients, only one patient died within 30 days 213 following the intervention due to a stroke. This patient underwent CEA in P2 and the 214 cause of death was a cancer. ) are really hard to understand.
Author Response
I have only some suggestions for the Auhtors:
- please provide a more descriptive methods section, better reporting how patients were enrolled during the different phases of the study; in tis present form M&M section is quite difficult to interoretate about this specific point – Comment accepted. The Methods section was rewritten. See Methods:
Consecutive patients who underwent CEA or CAS in a two time periods were included to the analysis. The first period was from January 2008 to December 2012 [P1] and the second period from January 2018 to December 2022 [P2]. Inclusion criteria were: 1/ ICA stenosis ≥70% detected using duplex sonography, computed tomography angiography or magnetic resonance angiography; 2/ indication for CEA or CAS according to valid guidelines7,27-29; 3/ age 40-85 years; signed Informed consent. Exclusion criterion was a failure to perform preoperative and postoperative cognitive testing due to aphasia, dementia, or unwillingness to undergo testing.
To avoid a selection bias, only the first 50 patients who underwent CEA and the first 50 patients who underwent CAS in each selected period and at the same time met the inclusion criteria were selected for the analysis. Finally, 267 out of 400 selected patients from the Ostrava University Hospital underwent cognitive testing and were analyzed.
- linguist revision (sse below) is mandatory – Comment accepted. The language editing will be done.
- please considere to discuss the following paper: Faraglia V, Palombo G, Stella N, Rizzo L, Taurino M, Bozzao A. Cerebral embolization during transcervical carotid stenting with flow reversal: a diffusion-weighted magnetic resonance study. Ann Vasc Surg. 2009 Jul-Aug;23(4):429-35. doi: 10.1016/j.avsg.2008.09.009. Epub 2008 Nov 28. PMID: 19041221. – Comment accepted. The article was cited. See Discussion:
If transfemoral approach represents a high risk for patient, transcervical CAS seems to be an option with good clinical outcome and reduction of postoperative DW-MRI ischemic lesions related with filter-protected CAS.42
- FARAGLIA, Vittorio, Giovanni PALOMBO, Nazzareno STELLA, et al. Cerebral embolization during transcervical carotid stenoting with flow reversal: a diffusion-weighted magnetic resonance study. Ann Vasc Surg. 2009, 23;(4):429-435.
Comments on the Quality of English Language
The whole manuscript requires an extensive linguistic revision, such as some sentences (like the following: Out of the 267 patients, only one patient died within 30 days 213 following the intervention due to a stroke. This patient underwent CEA in P2 and the 214 cause of death was a cancer. ) are really hard to understand. – Comment accepted. The language editing will be done.
Reviewer 2 Report
Comments and Suggestions for Authors
The authors present a study in which patients undergoing carotid endarterectomy (CEA) and carotid angioplasty with stenting (CAS) for treatment of carotid stenosis during the time intervals 2008-2012 and 2018-2022. The results showed positive change in cognitive function in patients treated in the later time point but not in the earlier time point. The results are clearly presented, and the discussion adequately interprets these results in context of prior investigations. Clarification of the methodology and text editing are required.
1. In the Methods section, it mentions 400 patients underwent CEA, from which 50 consecutive patients were considered. Among symptomatic patients, more patients were recruited from the second time point compared to the first time point. The reason for this is not clear. There is little explanation provided regarding how patients who underwent CAS were selected. Overall, more rigorous inclusion and exclusion criteria must be provided. A flowchart showing patient selection would be helpful.
2. The methods must be revised to improve clarity and readability.
For example, in the sentence “Out of a total of 400 patients who underwent carotid endarterectomy, the initial 50 consecutive patients in each group who had cognitive tests performed both before and after the intervention were selected” (Methods, lines 91-93) the reference to “each group” is not clear since no groups were defined in the methodology section at this point.
Methods, line 94: The terms “P1” and “P2” are used here without explanation until later. Instead, they should be explained at first use.
Methods 2.3, line 157: Reference #12 is cited as a study which used an MRI protocol which provided the basis of the MRI protocol of the current study. However, the cited article mentions using MRI without mentioning the specific protocol used. Therefore, citing this study seems to have limited value.
Comments on the Quality of English LanguageAbbreviations should be explained at first use and used consistently thereafter. For example, “CEA” is explained twice in the introduction section (lines 38 and 59). The full term “carotid endarterectomy” is used on line 50 and also line 105, when instead the abbreviation which was previously explained should be used.
Discussion, line 272: “ublished” should be “published”.
Author Response
The authors present a study in which patients undergoing carotid endarterectomy (CEA) and carotid angioplasty with stenting (CAS) for treatment of carotid stenosis during the time intervals 2008-2012 and 2018-2022. The results showed positive change in cognitive function in patients treated in the later time point but not in the earlier time point. The results are clearly presented, and the discussion adequately interprets these results in context of prior investigations. Clarification of the methodology and text editing are required.
- In the Methods section, it mentions 400 patients underwent CEA, from which 50 consecutive patients were considered. Among symptomatic patients, more patients were recruited from the second time point compared to the first time point. The reason for this is not clear. There is little explanation provided regarding how patients who underwent CAS were selected. Overall, more rigorous inclusion and exclusion criteria must be provided. A flowchart showing patient selection would be helpful. – Comment accepted. The Methods section was rewritten. See Methods:
Consecutive patients who underwent CEA or CAS in a two time periods were included to the analysis. The first period was from January 2008 to December 2012 [P1] and the second period from January 2018 to December 2022 [P2]. Inclusion criteria were: 1/ ICA stenosis ≥70% detected using duplex sonography, computed tomography angiography or magnetic resonance angiography; 2/ indication for CEA or CAS according to valid guidelines7,27-29; 3/ age 40-85 years; signed Informed consent. Exclusion criterion was a failure to perform preoperative and postoperative cognitive testing due to aphasia, dementia, or unwillingness to undergo testing.
To avoid a selection bias, only the first 50 patients who underwent CEA and the first 50 patients who underwent CAS in each selected period and at the same time met the inclusion criteria were selected for the analysis. Finally, 267 out of 400 selected patients from the Ostrava University Hospital underwent cognitive testing and were analyzed.
- The methods must be revised to improve clarity and readability.
For example, in the sentence “Out of a total of 400 patients who underwent carotid endarterectomy, the initial 50 consecutive patients in each group who had cognitive tests performed both before and after the intervention were selected” (Methods, lines 91-93) the reference to “each group” is not clear since no groups were defined in the methodology section at this point. – Comment accepted. The Methods section was rewritten. See Methods:
To avoid a selection bias, only the first 50 patients who underwent CEA and the first 50 patients who underwent CAS in each selected period and at the same time met the inclusion criteria were selected for the analysis. Finally, 267 out of 400 selected patients from the Ostrava University Hospital underwent cognitive testing and were analyzed.
Methods, line 94: The terms “P1” and “P2” are used here without explanation until later. Instead, they should be explained at first use. – Comment accepted. See Methods:
Consecutive patients who underwent CEA or CAS in a two time periods were included to the analysis. The first period was from January 2008 to December 2012 [P1] and the second period from January 2018 to December 2022 [P2].
Methods 2.3, line 157: Reference #12 is cited as a study which used an MRI protocol which provided the basis of the MRI protocol of the current study. However, the cited article mentions using MRI without mentioning the specific protocol used. Therefore, citing this study seems to have limited value. – Comment accepted. The MRI protocol is not relevant in this study. Thus, the section 2.3 (Magnetic resonance imaging) was deleted. See Methods.
Comments on the Quality of English Language
Abbreviations should be explained at first use and used consistently thereafter. For example, “CEA” is explained twice in the introduction section (lines 38 and 59). The full term “carotid endarterectomy” is used on line 50 and also line 105, when instead the abbreviation which was previously explained should be used. – Comment accepted.
Discussion, line 272: “ublished” should be “published”. – Comment accepted. The transcription error was corrected.
Round 2
Reviewer 2 Report
Comments and Suggestions for Authors
The authors have satisfactorily addressed all comments. No further comments are necessary.